# Learning nonlinear level sets for dimensionality reduction in function approximation

**Guannan Zhang**
Computer Science and Mathematics Division
Oak Ridge National Laboratory
zhangg@ornl.gov

**Jiaxin Zhang**
National Center for Computational Sciences
Oak Ridge National Laboratory
zhangj@ornl.gov

**Jacob Hinkle**
Computational Science and Engineering Division
Oak Ridge National Laboratory
hinklejd@ornl.gov

## Abstract

We developed a Nonlinear Level-set Learning (NLL) method for dimensionality reduction in high-dimensional function approximation with small data. This work is motivated by a variety of design tasks in real-world engineering applications, where practitioners would replace their computationally intensive physical models (e.g., high-resolution fluid simulators) with fast-to-evaluate predictive machine learning models, so as to accelerate the engineering design processes. There are two major challenges in constructing such predictive models: (a) high-dimensional inputs (e.g., many independent design parameters) and (b) small training data, generated by running extremely time-consuming simulations. Thus, reducing the input dimension is critical to alleviate the over-fitting issue caused by data insufficiency. Existing methods, including sliced inverse regression and active subspace approaches, reduce the input dimension by learning a linear coordinate transformation; our main contribution is to extend the transformation approach to a nonlinear regime. Specifically, we exploit reversible networks (RevNets) to learn nonlinear level sets of a high-dimensional function and parameterize its level sets in low-dimensional spaces. A new loss function was designed to utilize samples of the target functions' gradient to encourage the transformed function to be sensitive to only a few transformed coordinates. The NLL approach is demonstrated by applying it to three 2D functions and two 20D functions for showing the improved approximation accuracy with the use of nonlinear transformation, as well as to an 8D composite material design problem for optimizing the buckling-resistance performance of composite shells of rocket inter-stages.

## 1 Introduction

High-dimensional function approximation arises in a variety of engineering applications where scientists or engineers rely on accurate and fast-to-evaluate approximators to replace complex and time-consuming physical models (e.g. multiscale fluid models), so as to accelerate scientific discovery or engineering design/manufacture. In most of those applications, training and validation data need to be generated by running expensive simulations, that the amount of training data is often limited due to high cost of data generation (see §4.3 for an example). Thus, *this effort is motivated by the challenge imposed by high dimensionality and small data in the context of function approximation.*

One way to overcome the challenge is to develop dimensionality reduction methods that can build a transformation of the input space to increase the anisotropy of the input-output map. In this work, we assume that the function has a scalar output and the input consists of high-dimensional independent variables, such that there is no intrinsically low-dimensional structure of the input manifold. In this case, instead of analyzing the input or output manifold separately, we will learn low-dimensional structures of the target function's level sets to reduce the input dimension. Several methods have been developed for this purpose, including sliced inverse regression and active subspace methods. A literature review of those methods is given in §2.1. Despite many successful applications of those methods, their main drawback is that they use *linear* transformations to capture low-dimensional structures of level sets. For example, the existing methods for functions with linear level sets, e.g., $f(x) = \sin(x_1 + x_2)$ (the optimal linear transformation is a 45 degree rotation). When the level sets are nonlinear, e.g., $f(x) = \sin(\|x\|_2)$ (the spherical transformation is optimal), the number of active input dimensions cannot be reduced by linear transformations.

In this effort, we exploited reversible residual neural networks (RevNets) [6, 18] to learn the target functions' level sets and build nonlinear coordinate transformations to reduce the number of active input dimensions of the function. Reversible architectures have been developed in the literature [17, 19, 12] with a purpose of reducing memory usage in backward propagation, while we intend to exploit the reversibility to build *bijective* nonlinear transformations. Since the RevNet is used for a different purpose, we designed a new loss function for training the RevNets, such that a well-trained RevNet can capture the nonlinear geometry of the level sets. The key idea is to utilize samples of the function's gradient to promote the objective that most of the transformed coordinates move on the tangent planes of the target function, i.e., the transformed function is invariant with respect to those coordinates. In addition, we constrain the determinant of the Jacobian matrix of the transformation in order to enforce invertibility. The main *contributions* of this effort can be summarized as follows:

- Development of RevNet-based coordinate transformation model for capturing the geometry of level sets, which extends function dimensionality reduction to the nonlinear regime.

- Design of a new loss function that exploits gradient of the target function to successfully train the proposed RevNet-based nonlinear transformation.

- Demonstration of the performance of the proposed NLL method on a high-dimensional real-world composite material design problem for rocket inter-stage manufacture.

## 2   Problem formulation

We are interested in approximating a $d$-dimensional multivariate function of the form

$$y = f(\boldsymbol{x}), \quad \boldsymbol{x} \in \Omega \subset \mathbb{R}^d, \tag{1}$$

where $\Omega$ is a *bounded* domain in $\mathbb{R}^d$, the input $\boldsymbol{x} := (x_1, x_2, \ldots, x_d)^\top$ is a $d$-dimensional vector, and the output $y$ is a scalar value. $\Omega$ is equipped with a probability density function $\rho : \mathbb{R}^d \mapsto \mathbb{R}^+$, i.e.,

$$0 < \rho(\boldsymbol{x}) < \infty, \ \boldsymbol{x} \in \Omega \quad \text{and} \quad \rho(\boldsymbol{x}) = 0, \ \boldsymbol{x} \notin \Omega,$$

such that the manifold $\{\boldsymbol{x} | \boldsymbol{x} \sim \rho(\boldsymbol{x})\}$ does not have any intrinsically low-dimensional structure (e.g., $\rho$ is a uniform distribution in a $d$-dimensional hypercube). The target function $f$ is assumed to be first-order continuously differentiable, i.e., $f \in C^1(\Omega)$, and square-integrable with respect to the probability measure $\rho$, i.e., $\int_\Omega f^2(\boldsymbol{x})\rho(\boldsymbol{x})d\boldsymbol{x} < \infty$.

In many engineering applications, e.g., the composite shell design problem in §4.3, $f$ usually represents the input-output relationship of computationally expensive simulators. In order to accelerate a discovery/design process, practitioners seek to build an approximation of $f$, denoted by $\tilde{f}$, such that the error $f - \tilde{f}$ is smaller than a prescribed threshold $\varepsilon > 0$, i.e., $\|f(\boldsymbol{x}) - \tilde{f}(\boldsymbol{x})\|_{L^2_\rho(\Omega)} < \varepsilon$, where $\|\cdot\|_{L^2_\rho}$ is the $L^2$ norm under the probability measure $\rho$. As discussed in §1, the main challenge results from the concurrence of having high-dimensional input (i.e., large $d$) and small training data, which means the amount of training data is insufficient to overcome the curse of dimensionality. In this scenario, naive applications of existing approximation methods, e.g., sparse polynomials, kernel methods, neural networks (NN), etc., may lead to severe over-fitting. Therefore, our goal is to reduce the input dimension $d$ by transforming the original input vector $\boldsymbol{x}$ to a lower-dimensional vector $\boldsymbol{z}$, such that the transformed function can be accurately approximated with small data.

## 2.1 Related work

**Manifold learning for dimensionality reduction**. Manifold learning, including linear and nonlinear approaches [28, 27, 2, 14, 29, 26], focuses on reducing data dimension via learning intrinsically low-dimensional structures in the data. Nevertheless, since we assume the input vector $\boldsymbol{x}$ in Eq. (1) consists of independent components and the output $f$ is a scalar, no low-dimensional structure can be identified by separately analyzing the input and the output data. Thus, the standard manifold learning approaches are not applicable to the function dimensionality reduction problem under consideration.

**Sliced inverse regression (SIR)**. SIR is a statistical dimensionality reduction approach for the problem under consideration. In SIR, the input dimension is reduced by constructing/learning a *linear* coordinate transformation $\boldsymbol{z} = \mathbf{A}\boldsymbol{x}$, with the expectation that the output of the transformed function $y = h(\boldsymbol{z}) = h(\mathbf{A}\boldsymbol{x})$ is only sensitive to a very small number of the new coordinates of $\boldsymbol{z}$. The original version of SIR was developed in [23] and then improved extensively by [10, 24, 11, 9, 25]. To relax the elliptic assumption (e.g., Gaussian) of the data, kernel dimension reduction (KDR) was introduced in [15, 16]. Several recent work, including manifold learning with KDR [31] and localized SIR [30], were developed for classification problem. In §4, the SIR will be used to produce baseline results to compare with the performance of our *nonlinear* method.

**Active subspace (AS)**. The AS method [8, 7] shares the same motivation as SIR, i.e., reducing the input dimension of $f(\boldsymbol{x})$ by defining a linear transformation $\boldsymbol{z} = \mathbf{A}\boldsymbol{x}$. The main difference between AS and SIR is the way to construct the matrix $\mathbf{A}$. The AS method does not need the elliptic assumption needed for SIR but requires (approximate) gradient samples of $f(\boldsymbol{x})$ to build $\mathbf{A}$. For both SIR and AS, when the level sets of $f$ are nonlinear, e.g., $f(\boldsymbol{x}) = \sin(\|\boldsymbol{x}\|_2^2)$, the dimension cannot be effectively reduced using any linear transformation. An initial attempt of nonlinear AS method was conducted in [4] by analyzing local structures of isosurfaces, where its main drawback is the high online cost. The AS method will be used as another baseline to compare with our method in §4.

**Reversible neural networks**. We exploited the RevNets proposed in [6, 18] to define our nonlinear transformation for dimensionality reduction. Those RevNets describe bijective continuous dynamics while regular residual networks result in crossing and collapsing paths which correspond to non-bijective continuous dynamics [1, 6]. Recently, RevNets have been shown to produce competitive performance on discriminative tasks [17, 20] and generative tasks [12, 13, 21]. In particular, the non-linear independent component estimation (NICE) [12, 13] used RevNets to build nonlinear coordinate transformations to factorize high-dimensional density functions into products of independent 1D distributions. The main difference between NICE and our approach is that NICE seeks convergence in distribution (weak convergence) with the purpose of building an easy-to-sample distribution, and our approach seeks strong convergence as indicated by the norm $\|\cdot\|_{L_\rho^2}$ with the purpose of building an accurate pointwise approximation to the target function in a lower-dimensional input space.

## 3 Proposed method: Nonlinear Level sets Learning (NLL)

The goal of dimensionality reduction is to construct a *bijective nonlinear* transformation, denoted by

$$\boldsymbol{z} = \boldsymbol{g}(\boldsymbol{x}) \in \mathbb{R}^d \quad \text{and} \quad \boldsymbol{x} = \boldsymbol{g}^{-1}(\boldsymbol{z}), \tag{2}$$

where $\boldsymbol{z} = (z_1, \ldots, z_d)^\top$, such that the composite function $y = f \circ \boldsymbol{g}^{-1}(\boldsymbol{z})$ has a very small number of active input components. In other words, even though $\boldsymbol{z} \in \mathbb{R}^d$ is still defined in $\mathbb{R}^d$, the components of $\boldsymbol{z}$ can be split into two groups, i.e., $\boldsymbol{z} = (\boldsymbol{z}_{\text{act}}, \boldsymbol{z}_{\text{inact}})$ with $dim(\boldsymbol{z}_{\text{act}})$ much smaller than $d$, such that $f \circ \boldsymbol{g}^{-1}$ is only sensitive to the perturbation of $\boldsymbol{z}_{\text{act}}$. To this end, our method was inspired by the following observation:

**Observation:** *For a fixed pair $(\boldsymbol{x}, \boldsymbol{z})$ satisfying $\boldsymbol{z} = \boldsymbol{g}(\boldsymbol{x})$, if $\boldsymbol{x} = \boldsymbol{g}^{-1}(\boldsymbol{z})$, as a particle in $\Omega$, moves along a tangent direction, i.e., any direction perpendicular to $\nabla f(\boldsymbol{x})$, of the level set passing $f(\boldsymbol{x})$ under a perturbation of $z_i$ (the $i$-th component of $\boldsymbol{z}$), then the output of $f \circ \boldsymbol{g}^{-1}(\boldsymbol{z})$ does NOT change with $z_i$ in the neighbourhood of $\boldsymbol{z}$.*

Based on such observation, we intend to build and train a nonlinear transformation $\boldsymbol{g}$ with the objective that having a prescribed number of inactive components of $\boldsymbol{z}$ satisfy the above statement, and those inactive components will form $\boldsymbol{z}_{\text{inact}}$.

**Training data:** We need two types of data for training $\boldsymbol{g}$, i.e., samples of the function values and its gradients, denoted by

$$\Xi_{\text{train}} := \left\{ \left( \boldsymbol{x}^{(s)}, f(\boldsymbol{x}^{(s)}), \nabla f(\boldsymbol{x}^{(s)}) \right) : \ s = 1, \ldots, S \right\},$$

where $\{\boldsymbol{x}^{(s)} : s = 1, \ldots, S\}$ are drawn from $\rho(\boldsymbol{x})$, and $\nabla f(\boldsymbol{x}^{(s)})$ denotes the gradient of $f$ at $\boldsymbol{x}^{(s)}$. The gradient samples describe the tangent direction of the target function's level sets, i.e., the gradient direction is in perpendicular to all tangent directions. The requirement of gradient samples may limit the applicability of our approach to real-world applications in which gradient is not available. A detailed discussion on how to mitigate such disadvantage is given in §5.

## 3.1 The level sets learning model: RevNets

The first step is to define a model for the nonlinear transformation $\boldsymbol{g}$ in Eq. (2). In this effort, we utilize the nonlinear RevNet model proposed in [6, 18], defined by

$$\begin{cases} \boldsymbol{u}_{n+1} = \boldsymbol{u}_n + h \, \mathbf{K}_{n,1}^\top \, \boldsymbol{\sigma}(\mathbf{K}_{n,1} \boldsymbol{v}_n + \boldsymbol{b}_{n,1}), \\ \boldsymbol{v}_{n+1} = \boldsymbol{v}_n - h \, \mathbf{K}_{n,2}^\top \, \boldsymbol{\sigma}(\mathbf{K}_{n,2} \boldsymbol{u}_{n+1} + \boldsymbol{b}_{n,2}), \end{cases} \tag{3}$$

for $n = 0, 1, \ldots, N - 1$, where $\boldsymbol{u}_n$ and $\boldsymbol{v}_n$ are partitions of the states, $h$ is the "time step" scalar, $\mathbf{K}_{n,1}, \mathbf{K}_{n,2}$ are weight matrices, $\boldsymbol{b}_{n,1}, \boldsymbol{b}_{n,2}$ are biases, and $\boldsymbol{\sigma}$ is the activation function. Since $\boldsymbol{u}_n$, $\boldsymbol{v}_n$ can be explicitly calculated given $\boldsymbol{u}_{n+1}, \boldsymbol{v}_{n+1}$, the RevNet in Eq. (3) is reversible by definition. Even though our approach can incorporate any reversible architecture, we chose the model in Eq. (3) because it has been shown in [6] that this architecture has better nonlinear representability than other types of RevNets.

To define $\boldsymbol{g} : \boldsymbol{x} \mapsto \boldsymbol{z}$, we split the components of $\boldsymbol{x}$ evenly into $\boldsymbol{u}_0$ and $\boldsymbol{v}_0$, and split the components of $\boldsymbol{z}$ accordingly into $\boldsymbol{u}_N$ and $\boldsymbol{v}_N$, i.e.

$$\boldsymbol{x} := \begin{bmatrix} \boldsymbol{u}_0 \\ \boldsymbol{v}_0 \end{bmatrix} \quad \text{where} \quad \boldsymbol{u}_0 := (x_1, \ldots, x_{\lceil d/2 \rceil})^\top, \boldsymbol{v}_0 := (x_{\lceil d/2 \rceil + 1}, \ldots, x_d)^\top, \tag{4}$$

$$\boldsymbol{z} := \begin{bmatrix} \boldsymbol{u}_N \\ \boldsymbol{v}_N \end{bmatrix} \quad \text{where} \quad \boldsymbol{u}_N := (z_1, \ldots, z_{\lceil d/2 \rceil})^\top, \boldsymbol{v}_N := (z_{\lceil d/2 \rceil + 1}, \ldots, z_d)^\top, \tag{5}$$

such that the nonlinear transformation $\boldsymbol{g}$ is defined by the map $(\boldsymbol{u}_0, \boldsymbol{v}_0) \mapsto (\boldsymbol{u}_N, \boldsymbol{v}_N)$ from the input states of the $N$-layer RevNets in Eq. (3) to its output states, i.e.,

$$\boldsymbol{x} = \begin{bmatrix} \boldsymbol{u}_0 \\ \boldsymbol{v}_0 \end{bmatrix} \xrightarrow[\boldsymbol{g}^{-1}]{\boldsymbol{g}} \begin{bmatrix} \boldsymbol{u}_N \\ \boldsymbol{v}_N \end{bmatrix} = \boldsymbol{z}. \tag{6}$$

It was shown in [18] that the RevNet in Eq. (3) is guaranteed to be stable, so that we can use deep architectures to build a highly nonlinear transformation to capture the geometry of the level sets of $f$.

## 3.2 The loss function

The main novelty of this work is the design of the loss function for training the RevNet in Eq. (3). The new loss function includes two components. The first component is to inspired by our observation given at the beginning of §3. Specifically, guided by such observation, we write out the Jacobian matrix of the inverse transformation $\boldsymbol{g}^{-1} : \boldsymbol{z} \mapsto \boldsymbol{x}$ as

$$\mathbf{J}_{\boldsymbol{g}^{-1}}(\boldsymbol{z}) = [\boldsymbol{J}_1(\boldsymbol{z}), \boldsymbol{J}_2(\boldsymbol{z}), \ldots, \boldsymbol{J}_d(\boldsymbol{z})] \quad \text{with} \quad \boldsymbol{J}_i(\boldsymbol{z}) := \left( \frac{\partial x_1}{\partial z_i}(\boldsymbol{z}), \ldots, \frac{\partial x_d}{\partial z_i}(\boldsymbol{z}) \right)^\top \tag{7}$$

where the $i$-th column $\boldsymbol{J}_i$ describes the direction in which the particle $\boldsymbol{x}$ moves when perturbing $z_i$. As such, we can use $\boldsymbol{J}_i$ to mathematically rewrite our observation as: the output of $f(\boldsymbol{x})$ *does not change with a perturbation of $z_i$ in the neighborhood of $\boldsymbol{z}$, if*

$$\langle \boldsymbol{J}_i(\boldsymbol{z}), \nabla f(\boldsymbol{x}) \rangle = 0, \tag{8}$$

*where $\langle \cdot, \cdot \rangle$ denotes the inner product of two vectors.* The relation in Eq. (8) is illustrated in Figure 1. Therefore, the first component of the loss function, denoted by $L_1$, is defined by

$$L_1 := \sum_{s=1}^{S} \sum_{i=1}^{d} \left[ \omega_i \left\langle \frac{\boldsymbol{J}_i(\boldsymbol{z}^{(s)})}{\|\boldsymbol{J}_i(\boldsymbol{z}^{(s)})\|_2}, \nabla f(\boldsymbol{x}^{(s)}) \right\rangle \right]^2, \tag{9}$$

where $\omega_1, \omega_2, \ldots, \omega_d$ are user-defined *anisotropy weights* determining how strict the condition in Eq. (8) is enforced on each dimension. A extreme case could be $\boldsymbol{\omega} := (0, 1, 1, \ldots, 1)$, which means the objective is to train the transformation $\boldsymbol{g}$ such that the intrinsic dimension of $f \circ \boldsymbol{g}^{-1}(\boldsymbol{z})$ is one when $L_1 = 0$. Another extreme case is $\boldsymbol{\omega} = (0, \ldots, 0)$, which leads to $L_1 = 0$ and no dimensionality reduction will be performed. In practice, such weights $\boldsymbol{\omega}$ give us the flexibility to balance between training cost and reduction effect. It should be noted that we only normalize $\boldsymbol{J}_i$ in Eq (9), but not $\nabla f$, such that $L_1$ will not penalize too much in the regions where $\nabla f$ is very small. In particular, $L_1 = 0$ if $f$ is a constant function.

The second component of the loss function is designed to guarantee that the nonlinear transformation $\boldsymbol{g}$ is non-singular. It is observed in Eq. (9) that $L_1$ only affects the Jacobian columns $\boldsymbol{J}_i$ with $\omega_i \neq 0$, but has no control of the columns $\boldsymbol{J}_i$ with $\omega_i = 0$. To avoid the situation that the transformation $\boldsymbol{g}$ becomes singular during training, we define the second loss component $L_2$ as a quadratic penalty on the Jacobian determinant, i.e. ,

$$L_2 := (\det(\mathbf{J}_{g^{-1}}) - 1)^2, \tag{10}$$

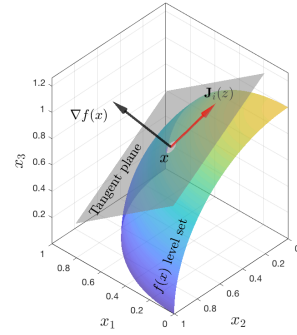

which will push the transformation to be non-singular and volume preserving. Note that $L_2$ can be viewed as a regularization term. In summary, the final loss function is defined by

$$L := L_1 + \lambda L_2, \tag{11}$$

where $\lambda$ is a user-specified constant to balance the two terms.

**Figure 1.** Illustration of the observation for defining the loss $L_1$ in Eq. (9), i.e., $f(\boldsymbol{x})$ is insensitive to perturbation of $z_i$ in the neighborhood of $\boldsymbol{z}$ if $\boldsymbol{J}_i(\boldsymbol{z}) \perp \nabla f(\boldsymbol{x})$, where $\boldsymbol{J}_i$ is defined in Eq. (7).

### 3.3 Implementation

The RevNet in Eq. (3) with the new loss function in Eq. (11) was implemented in PyTorch 1.1 and tested on a 2014 iMac Desktop with a 4 GHz Intel Core i7 CPU and 32 GB DDR3 memory. To make use of the automatic differentiation in PyTorch, we implemented a customized loss function in Pytorch, where the entries of the Jacobian matrix $\mathbf{J}_{g^{-1}}$ were computed using finite difference schemes, and the Jacobian $\det(\mathbf{J}_{g^{-1}})$ was approximately calculated using the PyTorch version of singular value decomposition. Since this effort focuses on proof of concept of the proposed methodology, the current implementation is not optimized in terms of computational efficiency.

## 4 Numerical experiments

We evaluated our method using three 2D functions in §4.1 for visualizing the nonlinear capability, two 20D functions in §4.2 for comparing our method with brute-force neural networks, SIR and AS methods, as well as a composite material design problem in §4.3 for demonstrating the potential impact of our method on real-world engineering problems. To generate baseline results, we used existing SIR and AS codes available at https://github.com/paulcon/active_subspaces and https://github.com/joshloyal/sliced, respectively. Source code for the proposed NLL method is available in the supplemental material.

### 4.1 Tests on two-dimensional functions

Here we applied our method to the following three 2-dimensional functions:

$$f_1(\boldsymbol{x}) = \frac{1}{2} \sin(2\pi(x_1 + x_2)) + 1 \quad \text{for } \boldsymbol{x} \in \Omega = [0, 1] \times [0, 1] \tag{12}$$

$$f_2(\boldsymbol{x}) = \exp(-(x_1 - 0.5)^2 - x_2^2) \quad \text{for } \boldsymbol{x} \in \Omega = [0, 1] \times [0, 1] \tag{13}$$

$$f_3(\boldsymbol{x}) = x_1^3 + x_2^3 + 0.2x_1 + 0.6x_2 \quad \text{for } \boldsymbol{x} \in \Omega = [-1, 1] \times [-1, 1]. \tag{14}$$

We used the same RevNet architecture for the three functions. Specifically, $\boldsymbol{u}$ and $\boldsymbol{v}$ in Eq. (3) were 1D variables (as the total dimension is 2); the number of layers was $N = 10$, i.e., 10 blocks of the form in Eq. (3) were connected; $\mathbf{K}_{n,1}, \mathbf{K}_{n,2}$ were $2 \times 1$ matrices; $\boldsymbol{b}_{n,1}, \boldsymbol{b}_{n,2}$ are 2D vectors; the activation function was $\tanh()$; the time step $h$ was set to 0.25; stochastic gradient descent method was used to train the RevNet with the learning rate being 0.01; no regularization was applied to the network parameters; the weights in Eq. (9) was set to $\boldsymbol{\omega} = (0, 1)$; $\lambda = 1$ in the loss function in Eq. (11); the training set included 121 uniformly distributed samples in $\Omega$, and the validation set included 2000 uniformly distributed samples in $\Omega$. We compared our method with either SIR or AS for each of the three functions.

The results for $f_1$, $f_2$, $f_3$ are shown in Figure 2. For $f_1$, it is known that the optimal transformation is a 45 degree rotation of the original coordinate system. The first row in Figure 2 shows that the trained RevNet can approximately recover the 45 degree rotation, which demonstrates that the NLL method can also recover linear transformation. The level sets of $f_2$ and $f_3$ are nonlinear, and the NLL method successfully captured such nonlinearity. In comparison, the performance of AS and SIR is worse than the NLL method because they can only perform linear transformation.

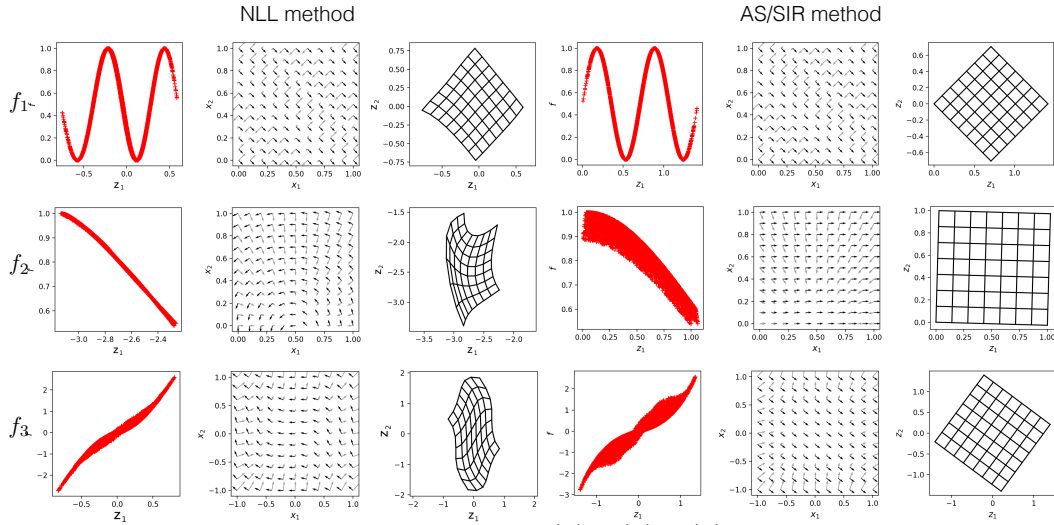

**Figure 2.** Comparison between NLL and AS/SIR for $f_1(\boldsymbol{x}), f_2(\boldsymbol{x}), f_3(\boldsymbol{x})$ in Eqs. (12)-(14) (rows 1-3 respectively). The first and fourth columns show the relationship between the function output and $z_1$, where the performance is better if the curve is thinner (i.e., the thickness of the cures shows the variation of $f \circ \boldsymbol{g}^{-1}$ w.r.t. $z_2$). The second and fifth columns show the gradient field (gray arrows) and the vector field of second Jabobian column $\boldsymbol{J}_2$, where the performance is better if the gray and black arrows are perpendicular to each other. The third and sixth columns show the transformation of a Cartesian mesh to the $\boldsymbol{z}$ space. Note that the AS method is shown for $f_1$, $f_3$ while the SIR method is shown for $f_2$; both methods were applied to all functions and showed very similar results. Since a linear transformation (45 degree rotation) is optimal in the case of $f_1$, both NLL and AS can learn such a transformation, but in the other cases the NLL method outperforms the linear methods.

## 4.2  Tests on 20-dimensional functions

Here we applied the new method to the following two 20-dimensional functions:

$$f_4(\boldsymbol{x}) = \sin\left(x_1^2 + x_2^2 + \cdots + x_{20}^2\right) \quad \text{and} \quad f_5(\boldsymbol{x}) = \prod_{i=1}^{20} \left(1.2^{-2} + x_i^2\right)^{-1} \qquad (15)$$

for $\boldsymbol{x} \in \Omega = [0, 1]^{20}$. We used one RevNet architecture for the two functions. Specifically, $\boldsymbol{u}$ and $\boldsymbol{v}$ in Eq. (3) were 10D variables, respectively; the number of layers is $N = 30$, i.e., 30 blocks of the form in Eq. (3) were connected; $\mathbf{K}_{n,1}, \mathbf{K}_{n,2}$ were $20 \times 10$ matrices; $\boldsymbol{b}_{n,1}, \boldsymbol{b}_{n,2}$ were 10-dimensional vectors; the activation function was $\tanh()$; the time step $h$ was set to 0.25; stochastic gradient descent method was used to train the RevNet with the learning rate being 0.05; $\lambda = 1$ for the loss function in Eq. (11); the training set includes 500 uniformly distributed samples in $\Omega$.

The effectiveness of the NLL method is shown as relative sensitivity indicators in Figure 3(a) for $f_4$ and Figure 3(b) for $f_5$. The sensitivity of each transformed variable $z_i$ is described by the *normalized* sample mean of the absolute values of the corresponding partial derivative. The definition of $\boldsymbol{w}$ in Eq. (9) provides the target anisotropy of the transformed function. For $f_4$, we set $\omega_1 = 0, \omega_i = 1$

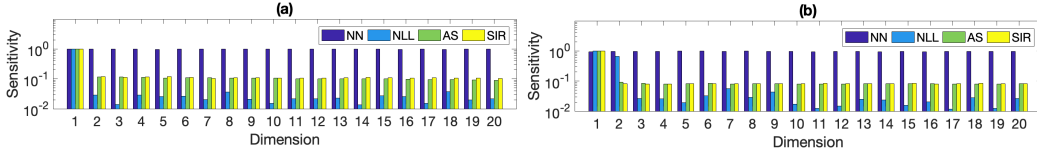

**Figure 3.** Comparison of relative sensitivities of the transformed function (a) $f_4 \circ \boldsymbol{g}^{-1}(\boldsymbol{z})$ and (b) $f_5 \circ \boldsymbol{g}^{-1}(\boldsymbol{z})$ with the original function and the transformed functions using AS and SIR methods.

for $i = 2, \ldots, 20$; for $f_5$ we set $\omega_1 = \omega_2 = 0$, $\omega_i = 1$ for $i = 3, \ldots, 20$. As expected, the NLL method successfully reduced the sensitivities of the inactive dimensions to two orders of magnitude smaller than the active dimensions. In comparison, the SIR and AS methods can only reduce their sensitivities by one order of magnitude using optimal linear transformations.

Next, we show how the NLL method improves the accuracy of the approximation of the transformed function $f \circ \boldsymbol{g}^{-1}$. We used two fully connected NNs to approximate the transformed functions, i.e., one has 2 hidden layers with 20+20 neurons, and the other has a single hidden layer with 10 neurons. The implementation of both networks was based on the neural network toolbox in Matlab 2017a. We used various sizes of training data: 100, 500, 10,000, and we used another 10,000 samples as validation data. All the samples are drawn uniformly in $\Omega$. The approximation error was computed as the relative root mean square error (RMSE) using the validation data. For comparison, we used the same data to run brute-force neural networks without any transformation, AS and SIR methods.

The results for $f_4$ and $f_5$ are shown in Table 1 and 2, respectively. For the 20+20 network, when the training data is too small (e.g., 100 samples), all the methods have the over-fitting issue; when the training data is very large (e.g., 10,000 samples), all the methods can achieve good accuracy [1]. Our method shows significant advantages over AS and SIR methods, when having relatively small training data, e.g., 500 training data, which is a common scenario in scientific and engineering applications. For the single hidden layer network with 10 neurons, we can see that the brute-force NN, AS and SIR cannot achieve good accuracy with 10,000 training data (no over-fitting), which means the network does not have sufficient expressive power to approximate the original function and the transformed functions using AS or SIR. In comparison, the NLL method still performs well as shown in Table 1(Right) and 2(Right). This means the dimensionality reduction has significantly simplified the target functions' structure, such that the transformed functions can be accurately approximated with smaller architectures to reduce the possibility of over-fitting.

Table 1: Relative RMSE for approximating $f_4$ in Eq. (15). (Left) 2 hidden layers fully-connected NN with 20+20 neurons; (Right) 1 hidden layer fully-connected NN with 10 neurons.

| | 100 data | | 500 data | | 10,000 data | | | 100 data | | 500 data | | 10,000 data | |
|---|---|---|---|---|---|---|---|---|---|---|---|---|---|
| | Valid | Train | Valid | Train | Valid | Train | | Valid | Train | Valid | Train | Valid | Train |
| NN | 96.74% | 0.01% | 61.22% | 1.01% | 9.17% | 7.72% | NN | 61.93% | 0.01% | 49.67% | 16.93% | 30.36% | 28.62% |
| NLL | 98.23% | 0.02% | 13.41% | 2.33% | 1.84% | 1.37% | NLL | 28.61% | 0.01% | 8.54% | 2.11% | 3.11% | 2.83% |
| AS | 95.42% | 0.03% | 65.98% | 1.09% | 2.36% | 1.81% | AS | 81.64% | 0.001% | 47.52% | 15.73% | 29.59% | 28.42% |
| SIR | 97.87% | 0.01% | 56.97% | 2.91% | 2.61% | 1.99% | SIR | 76.53% | 0.002% | 49.34% | 15.11% | 29.67% | 28.11% |

Table 2: Relative RMSE for approximating $f_5$ in Eq. (15). (Left) 2 hidden-layer fully-connected NN with 20+20 neurons; (Right) 1 hidden layer fully-connected NN with 10 neurons.

| | 100 data | | 500 data | | 10,000 data | | | 100 data | | 500 data | | 10,000 data | |
|---|---|---|---|---|---|---|---|---|---|---|---|---|---|
| | Valid | Train | Valid | Train | Valid | Train | | Valid | Train | Valid | Train | Valid | Train |
| NN | 40.95% | 0.005% | 33.92% | 11.10% | 3.56% | 4.14% | NN | 30.35% | 0.001% | 25.69% | 6.37% | 16.32% | 14.22% |
| NLL | 77.79% | 0.001% | 13.36% | 4.32% | 3.04% | 3.12% | NLL | 26.93% | 0.001% | 10.63% | 1.43% | 6.74% | 4.76% |
| AS | 66.64% | 0.002% | 39.73% | 3.38% | 6.21% | 3.32% | AS | 60.47% | 0.002% | 24.54% | 4.02% | 18.65% | 13.94% |
| SIR | 80.91% | 0.112% | 28.17% | 9.85% | 2.91% | 4.19% | SIR | 72.45% | 0.002% | 35.23% | 4.66% | 19.08% | 12.84% |

## 4.3 Design of composite shell for rocket inter-stages

Finally, we demonstrate the NLL method on a real-world composite material design problem. With high specific stiffness and strength, composite materials are increasingly being used for launch-vehicle structures. A series of large-scale composite tests for shell buckling knockdown factor conducted by NASA (see Figure 4(a)) aimed to develop and validate new analysis-based design guidelines for

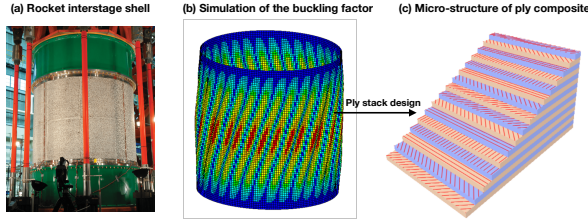

(a) Rocket interstage shell   (b) Simulation of the buckling factor   (c) Micro-structure of ply composite

Ply stack design

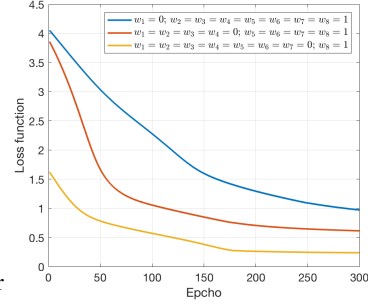

**Figure 4.** Illustration of the composite shell design problem for rocket inter-stages.

**Figure 5.** Loss function decay

safer and lighter space structure. Since the experimental cost is extremely high, numerical simulation, e.g., finite element method (FEM), is often employed to predict the shell buckling knockdown factor given a multi-layer ply stack design [5], as illustrated in Figure 4(c). The goal of this work is to implement an accurate approximation of this high-dimensional regression problem where the inputs are ply angles for 8 layers and the output is the knockdown factor which needs high precision for space structure design. However, the high fidelity FEM simulation is so time consuming that one analysis takes 10 hours and consequently it is impractical to collect a large data set for approximating the knockdown factor.

To demonstrate the applicability of our method to this problem, we used a simplified FEM model that runs relatively faster but preserves all the physical properties as the high-fidelity FEM model. As shown in Figure 4(b), a ply angle ranging from $0°$ to $22.5°$ that is assigned for each of 8 layers are considered in this example, i.e., the input domain is $\Omega = [0°, 22.5°]^8$. The RevNet has $N = 10$ layers; $\mathbf{K}_{n,1}, \mathbf{K}_{n,2}$ were $8 \times 4$ matrices; $\boldsymbol{b}_{n,1}, \boldsymbol{b}_{n,2}$ were 4-dimensional vectors; the activation function was $\tanh$; the time step $h$ was set to 0.1; stochastic gradient descent method was used with the learning rate being 0.05; $\lambda = 1$ for the loss function in Eq. (11).

Table 3: Relative sensitivities of the transformed functions for the composite material design model

| Method | Dim 1 | Dim 2 | Dim 3 | Dim 4 | Dim 5 | Dim 6 | Dim 7 | Dim 8 |
|---|---|---|---|---|---|---|---|---|
| Original | 1.0 | 0.85 | 0.72 | 0.61 | 0.51 | 0.45 | 0.39 | 0.36 |
| NLL | 0.68 | 1.0 | 0.12 | 0.011 | 0.075 | 0.036 | 0.024 | 0.018 |
| AS | 1.0 | 0.41 | 0.22 | 0.20 | 0.20 | 0.17 | 0.15 | 0.15 |
| SIR | 1.0 | 0.21 | 0.18 | 0.16 | 0.13 | 0.14 | 0.12 | 0.16 |

Like previous examples, we show the comparison of relative sensitivities in Table 3, where we allowed 3 active dimensions in the loss $L_1$, i.e. $\omega_1 = \omega_2 = \omega_3 = 0$, and $\omega_i = 1$ for $i = 4, \ldots, 8$.

As expected, the NLL method successfully reduced the input dimension by reducing the sensitivities of Dim 4-8 to two orders of magnitude smaller than the most active dimension, which outperforms the AS and SIR method. In Table 4, we show the RMSE of approximating the transformed function using a neural network with a single hidden layer having 20 neurons. The other settings are the same as the examples in §4.2. As expected, the NLL approach outperforms the AS and SIR in the small data regime, i.e., 500 training data.

Table 4: Relative RMSE for approximating the composite material design model.

| | 100 data | | 500 data | | 10,000 data | |
|---|---|---|---|---|---|---|
| | Valid | Train | Valid | Train | Valid | Train |
| NN | 65.74% | 0.01% | 67.57% | 24.77% | 3.74% | 3.52% |
| NLL | 63.18% | 0.02% | 11.96% | 5.13% | 2.51% | 2.17% |
| AS | 58.89% | 0.13% | 47.27% | 19.11% | 3.05% | 2.91% |
| SIR | 65.34% | 0.21% | 54.99% | 22.52% | 3.32% | 3.21% |

In Figure 5, we show the decay of the loss function for different choices of the anisotropy weights $\boldsymbol{\omega}$ in $L_1$, we can see that the more inactive/insensitive dimensions (more non-zero $\omega_i$), the slower the loss function decay.

## 5   Concluding remarks

We developed RevNet-based level sets learning method for dimensionality reduction in high-dimensional function approximation. With a custom-designed loss function, the RevNet-based nonlinear transformation can effectively learn the nonlinearity of the target function's level sets, so that the input dimension can be significantly reduced.

**Limitations**. Despite the successful applications of the NLL method shown in §4, we realize that there are several *limitations* with the current NLL algorithm, including (a) *The need for gradient samples.* Many engineering models do not provide gradient as an output. To use the current algorithm, we need to compute the gradients by finite difference or other perturbation methods, which will increase the computational cost. (b) *Non-uniqueness*. Unlike the AS and SIR method, the nonlinear transformation produced by the NLL method is not unique, which poses a challenge in the design of the RevNet architectures. (c) *High cost and low accuracy of computing Jacobians*. The main cost in the backward propagation lies in the computation of the Jacobian matrices and its determinant, which deteriorates the training efficiency and/or accuracy as we increase the depth of the RevNet.

**Future work**. There are several research directions we will pursue in the future. The first is to develop a gradient estimation approach that can approximately compute gradients needed by our approach. Specifically, we will exploit the contour regression method [22] and the manifold tangent learning approach [3], both of which have the potential to estimate gradients by using function samples. The second is to improve the computational efficiency of the training algorithm. Since our loss function is more complicated than standard loss functions, it will require extra effort to improve the efficiency of backward propagation.

## Acknowledgements

This material was based upon work supported by the U.S. Department of Energy, Office of Science, Office of Advanced Scientific Computing Research, Applied Mathematics program under contract ERKJ352; and by the Artificial Intelligence Initiative at the Oak Ridge National Laboratory (ORNL). ORNL is operated by UT-Battelle, LLC., for the U.S. Department of Energy under Contract DE-AC05-00OR22725.

## Footnotes

[1] 10% or smaller RMSE is considered as satisfactory accuracy in many engineering applications

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
