[Reviews · NeurIPS 2019]

Reviewer 1



The paper is well-writen, there are a few flaws in notation and typos though, such as: - (62) rho should map to \mathbb{R}_{\geq 0}, since it maps values to 0 - (133) remove paranthesis - (150) the partial derivatives are somewhat misleading, yet formal correct; i suggest you replace \frac{\partial x_1}{\partial z_i}(z) with \frac{\partial g^{-1}(z)}{\partial z_i}. By equation (6) it is clear, that this is meant, but i find this explicit formulation more comprehensible. - Figure 5: i think you mean epoch; the caption reads 'epcho' General comments: - All figures are too small. Without the ability to zoom, e.g. when you print out your paper, it is hardly possible to comprehend what is depicted. - the statement in line 153 and below is misleading and also wrong, if one does insist on formal correctness, equation 8 is a tautology: in Euclidean space every Orthogonal is a Perpendicular, and every Perpendicular is an Orthogonal, so this equivalence holds independent of the choice of x (assuming that z = g(x)); what you want to say is: f(x) does not change with z_i in the neighbourhood of z <=> \langle J_i(z), \nabla f(x) \rangle = 0 - Figure 2 is a bit hard to understand, though very expressive - in line 218 "sensitivity" is defined different than in Figure 2 - in equation 12 and 13 f_1 and f_2 are defined over [0,1] x [0,1] but in figure 2 we see that z_1 takes negative values - how is the domain of g being determined? Bottom line: I think this is a good and solid work which, except from the aforementioned flaws, supplies innovative methodology endowed with comprehensive thoughtful experiments, which do not only support the theoretical aspects of the methodology but also show the viability in real-world applications. #### After reading the rebuttal, I thank the authors for the clarifications and incorporating my suggestions.

Reviewer 2



The paper well structured, easy to read and technically sound. The problem under consideration is of importance in engineering and the proposed solution appears to be a first step towards a new general tool for more efficient function approximation. The use of RevNets in this context is an original and novel idea. On the other one could argue that the major novelty of this work is a bit shallow (a new problem specific loss function) and that, as the authors admit, use of the method is still restricted to rather specific scenarios. However, on the plus side, this paper makes headway into a direction of considerable industrial interest and demonstrates the potential of neural networks beyond their common use cases.

Reviewer 3



====================post rebuttal============================ After reading the authors' responses, I am quite satisfied with most of their explanations. In particular, the additional experiment on optimizing the dimensionality reduced functions for the real-world example looks quite persuasive, and the explanation about adding a dummy variable to address odd dimensional functions is also super valid. I also appreciate the authors for providing the detailed content of the modified paragraphs that they will include for the mathematical examples. The only small remaining issue is that for my point 6, the authors didn't seem to understand that the issue with Section 4.1 is that some of the sample points in the validation set may (almost) coincide with those in the training set, and the authors should make sure that they have excluded points that are sufficiently closed to the training set ones when generating the validation set, and clearly state this in the main text. That being said, I have decided to improve my score to 7 to acknowledge the sufficient improvement shown in the rebuttal. ========================================================== This paper considers the problem of dimensionality reduction for high dimensional function approximation with small data. Assuming no intrinsic low dimensional structure within the input space, the paper follows the level-set learning approaches in the literature, which exploits the low dimensional structure hidden in the dependency between the scalar output and high dimensional input (and in particular, the level sets), up to certain input variable transformation. The paper generalizes the traditional linear level-set learning approaches to the nonlinear regimes by introducing the RevNet structure as the model class for the nonlinear transform, and proposes a customized objective function based on the relation between level set low dimensionality and gradients, for learning a good nonlinear transform, which possesses the properties of being bijective and inactive to certain designated components of the transformed input variables. Some interesting experiments ranging from synthetic to real-world ones are also provided to demonstrate the efficiency of the proposed approach compared to the traditional ones. To the knowledge of the reviewer, the originality of this paper’s approach is beyond doubt. Although RevNet is an existing model structure, and although the proposed loss function is relatively straightforward given the motivation, the way that the authors combine these stuff, and the good performance of the proposed approach, still validate the significance of this paper. In addition, the paper is also generally well-written and easy to follow, with clear motivation, methodology description and experimental results. However, the reviewer also found the following issues to be addressed and/or places for improvement: 1. The description in lines 31-41 of why level-set approaches should be considered is not very well motivated. The authors may want to briefly mention the connection between inactive components of input variables, tangent spaces of level sets and gradient samples here, since otherwise the readers may need to wait until Section 3 before understanding the high level necessity, validity and mechanisms of level set approaches in terms of dimensionality reduction. 2. The authors may want to add a short paragraph (relatively early in the main text) on concrete examples of nonlinear dimensionality reduction, giving closed-form bijective transformation g that leads to low dimensional structures. The authors can simply provide two examples: 1) f(x)=\sin(\|x\|_2^2); 2) f(x)=\sin(x_1+x_2). The first example requires nonlinear transformation, which is polar/spherical transformation, and the second example allows for a simple 45-degree rotation. By clearly stating these examples together and comparing them, the readers can have a much better understanding of when and how linear and nonlinear transformation appear. 3. The mathematical setting in Section 2 is not very clear. Particularly, the definition of “independent” input variables is not formally given. This also makes it unclear how one could tell in practice whether the problem has independent or dependent input variables. In addition, the authors should clearly state where the density \rho comes from in practice (e.g., user-specified, as is done in the experiments section). 4. In Section 2.1, line 87, the function h is not clearly defined. Is h(Ax) actually \tilde{f}(x), i.e., the approximation of f(x)? Also, the authors should make it clearer whether they are using the most state-of-the-art versions of SIR and AS. In particular, is the github repo for SIR that the authors use as the benchmark using the more recent SIR with KDR? 5. In Section 3.1, the RevNet structure (3) seems to require that u_n and v_n have the same dimension, due to the transposes of K_{n,1} and K_{n,2}. And in particular, all the experiments conducted in this paper have even dimensions (2, 20 and 8). But how would one need to modify the RevNet structure when the dimension is odd? And how would this affect the results? 6. In the numerical experiments, the way that the authors generate the training and validation/test sets are not very valid. For example, in Section 4.1, some of the 2000 samples in the validation set may (almost) coincide with the 121 training points, especially given the small size of the regions. In Section 4.2, the validation set (for dimensionality reduction) is not mentioned. And more importantly, how is the training data (100, 500 and 10000) for the neural network function approximation related to the ones used for dimensionality reduction? In practice, given the expensiveness of function/gradient evaluations, shouldn’t they be the same, i.e., shouldn’t we reuse the same dataset when doing function approximation after the dimensionality reduction? 7. The explanations of the numerical experiments are not sufficiently clear. For example, for Section 4.1, Figure 2, the authors may want to mention clearly how one should tell from the plots which method is better. For columns 1 and 4, the performance is better if the curve is thinner, since thicker curves indicate multi-values caused by sensitivity to the hidden z_2. For columns 2 and 5, the key is to have the gray and black arrows perpendicular to each other. These things are not that clear for a first-time reader, and impede the fluency in reading. Btw, the authors may also want to mention what are the determinants of the learned transformation. And for Section 4.2, “normalized sample mean” and “partial derivative” are not clearly defined — what are the normalization and derivatives with respect to? And is there any activation in the neural network function approximation? 8. The authors may want to add an experiment on optimizing the dimensionality reduced functions for the real-world example in Section 4.3, and demonstrate the effect of dimensionality reduction in terms of optimization difficulty, as claimed in the abstract. Finally, some typos and minor suggestions: 1. Line 28: “simulations, that” -> “simulations, so that”. 2. The idea of RevNet and bijective neural network structures seems to be closely related to auto-encoders. The authors may want to slightly mention and explain the potential connection. 3. Lines 112-113: the authors may want to state that g is C^1, since later on Jacobians of g and g^{-1} are introduced. 4. Line 122: “objective that” -> “objective of”. Line 128: “is in perpendicular” -> “is perpendicular”. Line 148: “is to inspired” -> “is inspired”. Line 157: “A extreme” -> “An extreme”. 5. Lines 156-163: when explaining the motivation for the normalization, the authors may also want to clearly state that \nabla f can be zero, so cannot be normalized in general, but J_i are nonzero due to the non-singularity of J_{g^{-1}}. 6. Lines 166-168: why is singularity of J_{g^{-1}} related to zero entries in \omega? 7. Section 3.3: the authors may want to explain a bit more why finite differences and SVD are needed to utilize the automatic differentiation mechanism in PyTorch. 8. The dimensions of b seem to be wrong in Sections 4.2 and 4.3, since they should match the first dimension of K_{n,1} and K_{n,2}. In particular, in Line 213, b_{n,1} and b_{n,2} should be 20-dimensional; in Line 262, b_{n,1} and b_{n,2} should be 8-dimensional. 9. Line 219: “w” -> “\omega”. Line 233: “Table 1 and 2” -> “Tables 1 and 2”. Line 241: “Table” -> “Tables”.

[Author Response · NeurIPS 2019]

We are grateful to the reviewers for the insightful comments on our submission. Below we provide responses to
reviewer's major comments. All the minor comments will also be addressed in the revised manuscript.

**Reviewer 1**: "the statement in line 153 ...... in the neighbourhood of $z \iff \langle J_i(z), \nabla f(x) \rangle = 0$."
**Response**: We appreciate the reviewer's comment and suggestion. We will update line 153 to "*$f(x)$ does not change*
*with a perturbation of $z_i$ in the neighborhood of $z$, if and only if $\langle J_i(z), \nabla f(x) \rangle = 0$.*" Also, Eqn. (8) will be removed.

**Reviewer 1**: "in equation 12 and 13 $f_1$ and $f_2$ are defined ...... how is the domain of $g$ being determined?"
**Response**: The loss function does not impose constraints on the domain of $z$, which is why negative values of $z_1$ appear
in Figure 2. As $z$ does not have any physical meanings, it is unnecessary to force $z$ to be in a pre-determined domain
during the training. The domain of $z$ can be easily adjusted by translation and dilation after the training process.

**Reviewer 1**: "emphasize the need for gradient evaluations when you state the observation."
**Response**: The observation statement (line 118-120) will be updated to "*For a fixed pair $(x, z)$ satisfying $z = g(x)$, if*
*$x = g^{-1}(z)$ moves along a tangent direction, **i.e., any direction perpendicular to** $\nabla f(x)$, of the level set ......*".

**Reviewer 1**: "ambiguity of the notion of sensitivity (Figure 2 and below)". **Reviewer 4**: "...... for Section 4.1, Figure 2,
the authors may want to mention clearly how one should tell from the plots which method is better ....."
**Response**: We agree with both reviewers that the caption of Figure 2 is not very clear. The caption will be updated to
"*.......The first and fourth columns show the relationship between the output and $z_1$, **where the performance is better***
***if the curve is thinner (i.e., the thickness of the curves shows the variation of** $f \circ g^{-1}$ **w.r.t.** $z_2$**)**. The second and*
*fifth columns show the gradient field (gray arrows) and the vector field of second Jabobian column $J_2$, **where the***
***performance is better if the gray and black arrows are perpendicular to each other**. The third and sixth columns ...*"

**Reviewer 2**: "Are training times of the NLL low enough and its accuracy high enough to satisfy practitioners?".
**Reviewer 4**: "... add an experiment on optimizing the dimensionality reduced functions for the real-world example ... "
**Response**: To illustrate the significance of NLL to practitioners, we will add a new plot
in §4.3 to show the decay of the objective function for the optimal design in 3 cases:
(i) using the 8D FEM model, (ii) using the reduced 3D model, (iii) using the reduced
3D model + NN approximation. (i) v.s. (ii) shows the effectiveness of the NLL, i.e., the
3D optimization converges much faster than the 8D optimization; (ii) v.s. (iii) shows
that the NN approximation is accurate enough to exploit the dimensionality reduction
advantage. In addition, Case (iii) is much faster than Case (ii) because evaluating the
NN is very efficient compared to the evaluating the FEM model in Case (ii).

**Reviewer 4**: "2. The authors may want to add a short paragraph (relatively early in the main text) on ......"
**Response**: We appreciate the reviewer's suggestion and will add the suggested examples to line 40 as "*... structures of*
*level sets. For example, the existing methods for functions with linear level sets, e.g., $f(x) = \sin(x_1 + x_2)$ (the optimal*
*linear transformation is a $45°$ rotation). When the level sets are nonlinear, e.g., $f(x) = \sin(\|x\|^2)$ (the spherical*
*transformation is optimal), the number of active input dimensions cannot be reduced by linear transformations.*"

**Reviewer 4**: "3. The mathematical setting in Section 2 is not very clear. ......"
**Response**: The independence assumption was used to emphasize that our method can deal with functions that have no
intrinsic low-D structure in the input space. As independence is not a necessary condition for our method to work, we
will remove such assumption, as well as make it clear that $\rho(x)$ is a user-specified distribution in the revised manuscript.

**Reviewer 4**: "4. In Section 2.1, line 87, the function $h$ is not clearly defined ......"
**Response**: (i) $h$ is an implicitly defined link function mapping from $z$ to $y$, the composition of $h$ and $A$ is an
approximation of $f$. (ii) The AS and SIR codes used for comparison to the NLL represent the state of the art of both
methods. We did not use the SIR code with KDR, because the use of KDR will lead to *irreversible transformations* (i.e.,
$g^{-1}(z)$ may not exist), such that the relationship between $z$ and $y$ may not be a function, i.e., one value of $z$ may be
associated with multiple $y$ values. Those explanations will be made clear in the revised manuscript.

**Reviewer 4**: "5. In Section 3.1, the RevNet seems to require that $u_n$ and $v_n$ have the same dimension, ......"
**Response**: The used RevNet does require that $u_n$ and $v_n$ have the same dimension. When the dimension of $x$ is odd,
we can rewrite/extend $f(x)$ to $f(x, x^*)$ by adding one dummy variable $x^*$. Since $y$ does not really depend on $x^*$, we
will have $\partial f / \partial x^* = 0$, such that the loss function does not impose any constraint on $\partial x^* / \partial z_i$ (the only constraint on
$x^*$ is imposed by the regularizer $L_2$). Even though the $d + 1$-dimensional problem is generally harder to solve than
the original $d$-dimensional problem, we do not think it will significantly affect the performance of NLL because the
extension $f(x, x^*)$ is not sensitive with respect to $x^*$ from the beginning.

**Reviewer 4**: "6. ...... the way that the authors generate the training and validation/test sets are not very valid......"
**Response**: (i) §4.1 is to visualize the nonlinear capability of the NLL approach. To this end, we intended to remove
any over-fitting effect by using a dense training set for clear illustration in the first and fourth columns in Figure 2.
(ii) §4.2 is to show how the NLL helps alleviate the over-fitting issue, where the validation set with $10,000$ samples
were generated uniformly in $\Omega$. (iii) The training set for dimensionality reduction was reused in approximating the
transformed function $z \mapsto y$ using neural networks.

**Reviewer 4**: "7. ...... what are the normalization and derivatives w.r.t.? Is there any activation in the NN approximation?"
**Response**: The normalization constant is the maximum sensitivity index, such that the biggest sensitivity value is one in
Figure 3. The partial derivatives are defined by $\partial y / \partial z_i$ for $i = 1, \ldots, d$. Also, $\tanh()$ is used as the activation for NN.

[Meta-Review · NeurIPS 2019]

The paper proposes an interesting dimensionality reduction method for function approximation by generalizing linear level set learning methods to non linear level sets using the RevNet model structure and by introducing a loss function designed to give preference to functions that are sensitive only to few non linear coordinates. The paper is well-written and easy to understand. The methodology is clearly described and the experimental results are convincing. Hence, we recommend acceptance.